# *Trypanosoma cruzi* Genome 15 Years Later: What Has Been Accomplished?

**DOI:** 10.3390/tropicalmed5030129

**Published:** 2020-08-06

**Authors:** Jose Luis Ramirez

**Affiliations:** Instituto de Estudios Avanzados, Caracas, Venezuela and Universidad Central de Venezuela, Caracas 1080, Venezuela; ramjoseluis@gmail.com

**Keywords:** *Trypanosoma cruzi*, genome, parasite, neglected diseases

## Abstract

On 15 July 2020 was the 15th anniversary of the *Science* Magazine issue that reported three trypanosomatid genomes, namely *Leishmania major*, *Trypanosoma brucei*, and *Trypanosoma cruzi*. That publication was a milestone for the research community working with trypanosomatids, even more so, when considering that the first draft of the human genome was published only four years earlier after 15 years of research. Although nowadays, genome sequencing has become commonplace, the work done by researchers before that publication represented a huge challenge and a good example of international cooperation. Research in neglected diseases often faces obstacles, not only because of the unique characteristics of each biological model but also due to the lower funds the research projects receive. In the case of *Trypanosoma cruzi* the etiologic agent of Chagas disease, the first genome draft published in 2005 was not complete, and even after the implementation of more advanced sequencing strategies, to this date no final chromosomal map is available. However, the first genome draft enabled researchers to pick genes a la carte, produce proteins in vitro for immunological studies, and predict drug targets for the treatment of the disease or to be used in PCR diagnostic protocols. Besides, the analysis of the *T. cruzi* genome is revealing unique features about its organization and dynamics. In this work, I briefly summarize the actions of Latin American researchers that contributed to the first publication of the *T. cruzi* genome and discuss some features of the genome that may help to understand the parasite’s robustness and adaptive capabilities.

## 1. Chagas, the Disease

Chagas disease or American trypanosomiasis is considered a zoonosis caused by the flagellate *Trypanosoma cruzi*. The parasite has a wide range of mammalian hosts and insect vectors of the Triatominae family. According to the WHO [1,2], there are around 6 to 10 million people affected by the disease and 12,000 deaths per year, and due to migration, the disease is now present in non-endemic countries of Europe, Canada, and Australia. Infected patients can transition from an acute infection to a chronic and often lethal phase of the disease. Hitherto, however, the only drugs widely used to treat the disease are nifurtimox and benznidazole, which are very effective, when given at the early stages of the infection. Non-vectorially transmitted infections, such as oral or congenital, present new challenges for treatment [3], and the great variability of *T. cruzi* surface antigens has precluded the development of effective vaccination.

*Trypanosoma cruzi* populations are complex and using molecular markers seven distinct taxonomic units (DTUs) [4] have been reported, out of which two, namely DTU V and VI are hybrids. However, its population dynamic is currently a subject of debate [5].

## 2. The Years before the 2005 Publication, a Brief Description of Latin America’s (LA) Participation in the Sequencing of *the Trypanosoma cruzi* Genome

When the sequencing of the human genome [6] was proposed, funding agencies and the public showed serious concerns about the sizable funds dedicated to this task when more urgent needs were not properly funded [7,8,9]. These concerns, when applied to the genome of pathogens causing neglected diseases, are magnified. For basic researchers working on Chagas disease, it was evident that despite many years of biochemical, clinical, epidemiological and immunological studies, no new tools for the diagnostic or treatment of the disease were available, and the successful vector control measurements implemented in Latin America South-Cone countries were not as successful when applied to sylvatic environments where Reduvii bugs are not limited to domestic environments [10]. Hence, what makes *Trypanosoma cruzi* such a difficult organism to control or eliminate? Is the key to answering all these problems in its genome? 

Without sharing the reductionist view held by some proponents of the Human Genome Project Initiative (HGP) [6,11], that the key to understanding all genetic diseases was in the genome, LA researchers decided to launch a *Trypanosoma cruzi* genome initiative [12] hoping to piggyback on the successful Brazilian *Xyllela fastidiosa* genome project [13]. This initiative was justified, having in mind the following arguments: the training of young LA scientists in the emerging technologies of genomics and bioinformatics and preparing them to take advantage of public databases to foster innovation. Second, to achieve a better understanding of this parasite whose infestations negatively impact LA society. Third, to facilitate molecular biology studies in *T. cruzi* by providing a draft sequence that any laboratory could use to develop diagnostic tools or discover new therapeutic targets. Fourth, to reach out and unite efforts of researchers working on Chagas. 

In 1993, Dr. Mariano Levin of INGEBI, Buenos Aires, proposed one of the first genome projects for *T. cruzi* to the Biotechnology Program III of the Ibero American Science and Technology Program (CYTED). CYTED sponsored meetings bringing together molecular biologists from Spain, Sweden, and LA to design a strategy to sequence the genome [10,14]. Later, the WHO made a call for trypanosomatid genome research, and in 1994 on the premises of the Fundacao Oswaldo Cruz in Rio de Janeiro, Brazil, the most important meeting on Trypanosomatidae genome research took place [15]. For the occasion researchers from all over the world working on trypanosomatids converged to discuss a genome sequencing strategy for some of the most important pathogens of the Trypanosomatidae family, namely *Leishmania*, *Trypanosoma brucei,* and *Trypanosoma cruzi.* As an outcome, *Leishmania major*, *Trypanosoma brucei,* and *Trypanosoma cruzi* strain CL Brener (CLB) were selected as model species for genome projects. Furthermore, a more extended network led by Carlos Frasch of San Martin University, Argentina, was created including researchers from Sweden, the US, the UK, and France [16]. The approach to sequence the *T. cruzi* genome was the chromosome-by-chromosome or map-as-you-go strategy of the Human Genome Project (HGP) [6]. The work was divided among participant laboratories using the limited funding of local agencies. In three years, the laboratories constructed genome libraries either from DNA (genomic libraries) or mRNA (expression libraries). The work of these early years generated important information about the CLB genome, such as (i) the electrokaryotype profile revealing a large number of chromosomal bands and homologous chromosomes presenting size polymorphisms, and extra-numbers (aneuploidy) [17,18]; (ii) large non-syntenic zones of the genome containing multigene families e.g., the trans-sialidases gene family [19]; (iii) numerous putative transposons [20,21]; (iv) the mapping and expression of important infectivity genes; (v) and that *T. cruzi* telomeres and subtelomeres were enriched in transposon sequences and members of other large gene families [22]. When these actions were well underway, the initiative was confronted with the hard reality of the DNA sequencing costs at that time. The cost of the 3 × 10^9^ bp human genome was calculated to be between 500 million to 1 billion USD. Taking as a reference the HGP lower estimate, for the CLB ~6 × 10^7^ bp genome the approximate cost was 20 million USD. Fortunately for *L. major* and *T. brucei* the Welcome Trust provided support for these projects, whereas *T. cruzi* received no funding. A simplistic approach for *T. cruzi* was to develop end-linking maps with bacterial artificial chromosomes (BACs) or Yeast artificial chromosomes (YACS) recombinants (and assume that, given the taxonomical proximity to *T. brucei,* the gaps could be closed using the *T. brucei* synteny information. The work for the CLB genome continued using the original strategy until Dr. Najib El Sayed of the Institute for Genomic Research (TIGR), teaming up with colleagues of the Karolinska Institute (Sweden) and Seattle Biomedical Research Institute (USA), sequenced the whole *T. cruzi* genome using the Whole Genome Shotgun (WGS) approach developed by Venter [11] for the alternative human genome project. This approach consists of the random cloning of DNA fragments to generate libraries with different fragment sizes, and then carry out an automated computer assembly. Through WGS, the first draft of the CLB genome was accomplished in record time, the rest of the participant laboratories engaged in gene annotation and mapping their contigs and clones on this raw draft.

## 3. A Brief Description of the Initial Findings in the CLB Genome

With the 2005 publication of the first draft [23], scientists realized how complex the *T. cruzi* genome was; the automated annotations revealed a much higher degree of sequence repetitions, including large multigene families, out of which the Mucin Associated Surface Protein (MASP) family was detected for the first time. Repeated sequences comprised about half of the total genome. These repetitions often caused the collapse of the automatic assembling. Even though the CLB genome was diploid, it turned out to be a hybrid of two *T. cruzi* strains, namely Esmeraldo and non-Esmeraldo. In the current taxonomical classification, CLB is included in the hybrid DTU VI [4,23]. Despite all these obstacles, syntenic groups were assembled, confirming previous studies [19] about the lack of synteny for some surface protein genes. Although a final assembly was not achieved, the genome draft presented non-syntenic blocks or islands containing highly repeated genes and blocks made up of simple genes interrupted by a few copies of retrotransposons and genes from surface protein families. At the start of each syntenic block, there were breakpoints made up of simple repeated sequences being the origins of the bidirectional polycistronic transcription.

Similar to *T. brucei*, *T. cruzi* had a large number of transposons and transposon-associated sequences, but in *T. cruzi* the non-LTR transposons have the potential to be functional [24,25], and there is no RNAi machinery to control their activities.

The subtelomeres were confirmed as complex and different among all Trypanosmatids [24,26], and in the case of CLB they were non-syntenic and often enriched in highly repeated sequences.

A comparative analysis with the *L. major* and *T. brucei* genomes revealed that 32% of *T. cruzi* proteins were species-specific [23]. Another interesting feature was the expansion of the kinome (body of kinases), and the presence of enzymes necessary for meiotic recombination. Surprisingly, important genes for end joining non-homologous recombination (EJNHR) were missing in the three typanosomatids sequenced [24]. One the major contributions of the 2005 genome draft was that for the first time a large scale integration of *T. cruzi* metabolism networks was made possible, which opened the way for whole proteome analysis [27,28] and the application of system biology approaches [29].

## 4. What is New and What is Challenging

Even today, despite the refinements of new sequencing and bioinformatics technologies we still do not have a solid *T. cruzi* chromosomal map and considering the heterogeneity and genome variability of *T. cruzi* populations, we might never have one. Perhaps, a less noisy view of the *T. cruzi* genome could be obtained by single-cell genome sequencing [30,31,32]. So far, it seems that the best chromosomal map assembled for CLB was obtained by Weatherly et al. (2009) [33] using the information from the parental strains of the two haplotypes that make up the CLB genome, and closing the gaps with the synteny information derived from other trypanosomes, and/or BAC end joining. These authors defined 41 pairs of chromosomes as well as two artificially assembled contigs that could not be assigned at any chromosome. More recently, through long sequence reads obtained by the PACBio technology several laboratories have succeeded in closing some of the gaps in the original genome draft [30,34,35,36,37,38].

Despite not having a firm chromosomal map, the availability of the original genome blueprint, plus the speed and affordability of current DNA sequencing techniques, the genomes from all DTUs (distinct taxonomic units or near-clades) have been sequenced, and population analysis is now being done with whole genomes instead of isolated makers. The comparative analyses of *T. cruzi* isolates have shown variations in gene copy numbers and aneuploidy not only between the DTUs but also within DTUs from sympatric geographic areas [30,39]. Using the same tools, sexual recombination and parasexual hybrid formation have been detected in some *T. cruzi* populations [37,39,40,41,42].

Aneuploidy and variable gene copy numbers of genes [30,42,43] are at the base of the expansion of important genes for infecting and evading the host’s immune system, such as surface proteins, glycosidases, amastins, and GP63 proteases [44,45]. Aneuploidy, a condition that in other eukaryotes causes cell growth retardation [46], does not seem problematic for *T. cruzi,* and likewise, in *Candida glabrata* [47] an increase in gene dosage may serve as a dynamic mechanism to generate variability for adaptive purposes. Although there is the possibility that *T. cruzi* aneuploids are laboratory artifacts, aneuploids have been found in clones from non-aneuploid sympatric populations [30].

Copy Number Variations [3,34,35,37,38,44,48,49,50] originated by gene duplication, ectopic recombination, unequal-crossing-over, gene mobilization via transposons, chromosome duplications, and others, may contribute to variants with a selective advantage. As stated before, *T. cruzi*’s genes coding for surface proteins are notably expanded [30,34,35,36,49,50] and some variants have already been adapted to processes related to the infectivity and interaction with the host [51,52,53,54]. In addition to complete genes, the *T. cruzi* genome harbors an enormous repertoire of pseudogenes, which could be seen as a futile waste of energy and are likely to be reservoirs for the generation of gene variants. However, another possibility is that they [55,56,57,58] act as transcription regulators [59]. One of the keys to the understanding of *T. cruzi* genome dynamics might reside in the role that pseudogenes might play. As mentioned, the participation of transposons and repeated sequences in shaping the *T. cruzi* genome is supported by their presence in regions where gene synteny is broken and polycistronic transcription starts. Furthermore, it has been suggested that transposon may provide these sites [24,60].

The characteristics of the *T. cruzi* genome raises some questions: how can the parasite keep its very precise form and differentiation program with a very variable genome, including many non-syntenic groups of genes? How can mitotic and meiotic mechanisms efficiently operate with extra chromosomes and non-syntenic chromosomal blocks? Finally, how can transcription processes be regulated in rearranged syntenic blocks? 

The block structure of the *T. cruzi* genome allows the parasite to preserve in syntenic groups of polycistronic transcription the core functions (housekeeping genes) necessary for the parasite differentiation, metabolism, and division, whereas the blocks containing genes for important surface determinants display constant reshuffling generating variability. These two blocks appear to have different evolution patterns [23,34,36,37].

The generation of variability through non-sexual processes such as ectopic recombination has been experimentally assessed between *T. cruzi* subtelomeres [61], and given the large sequence variability registered at the subtelomeres, together with their enrichment in contingency genes such as the transialidases; transposons and transposons associated sequences seem to be important locations for the generation of variability [22,37,62,63]. The mobilization of variants may occur either by ectopic recombination and/or mediated by a transposon (gene transduction) [64,65]. In all these events homologous recombination (HR) seems to play an important role, and perhaps targeting some of the enzymes involved in HR may help to solve some of these questions.

Considering sexuality and the occurrence of meiotic processes, most genes necessary for meiotic recombination are present in *T. cruzi* [23], but a cytological study of meiosis is precluded, not only by the small size of the chromosomes but also by its lack of condensation and non-dissolution of the nuclear membrane during division processes. Therefore, pieces of evidence for hybridization and sexual exchanges come from the molecular mapping of genetic markers [43] and, more recently, through whole-genome analysis [30,34,39,40,41,42]. With this type of approach, researchers have accrued pieces of evidence of sexual recombination in *T. cruzi* populations from Brazil, Peru, Ecuador, and Colombia. 

A different aspect of sexual or parasexual exchanges in *T. cruzi* has to do with the inheritance of mitochondrial DNA. The kinetoplast, a DNA body common to all Kinetoplastida, is an intricate mesh of mini and maxicircles; the former participate in the editing process of the encrypted genes encoded in the maxicircle, which is the equivalent of the mitochondrial DNA in other organisms. There are claims of the uniparental contribution of K-DNA in parasexual hybrids [23,66] but with pieces of evidence of exchanges with other clades or DTUs. Little is known about how these inter-DTUs introgression processes occur [30,67,68].

Many questions about sexuality remain, such as when and where sexual recombination occurs? Are gametes ever produced? Are both gametes contributing with mitochondrial DNA, in which differentiation forms, and in which host does it take place?

## 5. Clonal or Not Clonal, is That the Question?

A panmictic evolution with sexual exchanges through gametes or parasexual fusion of cells [5,69] does not appear to contradict the general hypothesis that the *T. cruzi* population structure appears to be clonal [69,70], or as more recently coined, has a predominantly clonal signal [70]. Either way, the apparent infrequency of sexual exchanges indicates it is not obligatory for *T. cruzi.* Clonality can result from genome instability since aneuplody and variable gene copy numbers could be obstacles for efficient meiotic recombination. However, under challenging situations, this instability can generate gene variants with adaptive advantages at a faster rate when compared to sexual recombination. This fact agrees with the low frequency of sexual recombination registered in whole genomic analysis or the limited recombination between hybrid strain chromosomes [23,34,70]. Sexual recombination has been detected mainly in closely related populations, and rarely between strains from different near-clades. Some authors claim that sexual recombination occurs more frequently but remains undetected because it happens in closely related parasites (endogamy) [40]. The key to alternatives ways of evolution may reside in the heterogeneity of *T. cruzi* populations, which even under in vitro culturing are very heterogeneous [71,72]. Within the population, some members may retain the ability for sexual exchanges, while the majority stays clonal and adapted to a given environment. The accumulation of deleterious mutations during clonal propagation may drive a cell to its death and be eliminated from the population (Muller Rachet) [73]. However, since *T. cruzi* populations do not divide synchronously, not all cells will reach this end simultaneously, and some of the survivors with fewer genome alterations or those that have reversed to the diploid [30] condition by chromosomal loss, can engage in sexual or parasexual exchanges to replenish population fitness. Although it has been suggested that sexual recombination occurs inside the vertebrate host [68], it seems to the parasite’s advantage to be able to generate variants at all stages of differentiation, as well as in all hosts, so populations can go either way, panmictic or clonal.

Genome-wise, *T. cruzi* seems to be the master of variation, a property that explains its resilience, its capacity to infect a wide range of hosts and vectors, the utilization of multiple ways of infection, its capacity to recover its karyotype after a massive Gamma ray irradiations [74], and the rapid acquisition of surface protein variants to escape the host’s immune system, a fact that makes the attainment of an effective vaccine very difficult.

## 6. Final Considerations

One indisputable fact is that despite the many genetic variations and polymorphisms recorded among *T. cruzi* strains, the parasite is a pathogenic entity that keeps its basic cell architecture and differentiation programs and infects humans causing Chagas disease. For practitioners, infections by *T. cruzi,* regardless of the considerations about the parasite types, are treated in the same classical way, which involves the drugs nifurtimox or benznidazol.

Nonetheless, the information provided by the 2005 publication paved the way for the generation of new diagnostic tools, the discovery of important metabolic routes, the publication of nearly 2000 research papers, and the production of a wealth of information that allows a more systemic approach to *T. cruzi* biology and a large scale search for druggable targets [75,76,77].

The genetic experimentation with *T. cruzi* has lagged behind *Leishmania* and *T. brucei* due to the lack of cloning vectors with inducible promoters and/or an RNAi machinery that could allow for checking the importance and function of genes, and although the recent use of CRISPR technologies has facilitated this task [78,79], we are far from what is known about *T. brucei.* A more in-depth insight into the causes of *T. cruzi* genome plasticity and genetic variability may lead to understanding how the parasite generates diversity and provide us with better tools to effectively tackle Chagas disease.

Last, but certainly not least, the Trypanosomatid genomes contributed to the generation of a new cohort of young LA researchers in genomics and bioinformatics, some of whom are currently successfully engaged in fighting other diseases.

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
