# Peer review of "Trypanosoma cruzi Genome 15 Years Later: What Has Been Accomplished?"

_tropicalmed, 2020, doi:10.3390/tropicalmed5030129_

Round 1
Reviewer 1 Report
Ramirez resents a review of progress made T. cruzi biology since the publication of the genome. The review initially focuses on the contribution of Latin American researchers to the development of the genome project. Subsequently, it details some of the insights gained through analysis of the original CL Brener sequence and other more recent sequencing efforts and highlights an extensive list of questions still to be addressed by researchers.
Ramirez’s review is informative and, overall, well written. I’ve made some minor comments, below, and highlighted a selection of typos and grammatical issues that may cause confusion – the latter is not an exhaustive list, however, and the manuscript could benefit from some close proof reading.
Line 49 – ‘…sylvatic environments where Reduvii bugs are not domiciliated.’. I’m unclear what this means. Maybe: ‘…sylvatic environments where Reduvid bugs are not limited to domestic environments’
Line 122 – ‘genes for the non-homologous end joining DNA repair pathway were missing’. Note, these are also missing from Leishmania and T. brucei (situation for all three trypanosomatid species reviewed in Genois et al 2014; DOI: 10.1128/MMBR.00045-13)
Line 149 – alternative (better?) examples are the sub-telomeric aneuploidy and chromosomal aneuploidy seen in T. brucei and Leishmania, respectively.
Lines 175/183 – are the more variable genes, such as the surface proteins, highlighted earlier, present at sub-telomeres? Contingency genes tend to be located at sub-telomeres, e.g. in T. brucei and Plasmodium. This point should be more clearly made here – as it is, it’s unclear why sub-telomeres are under discussion.
Line 205/206 – ‘…either way of evolution does not seem to be obligatory for T. cruzi’. This is an odd construction, which needs rephrasing. May be something like: ‘…signal [70]. Either way, the apparent infrequency of genetic exchange indicates that it is not obligatory for T. cruzi.’
Line 86 – ‘bp’
Line 94 – should be ‘…Whole Genome Shotgun (WGS)…’
Line 104 – replace ‘comprehended’ with ‘comprised’
Line 106 – ‘Esmeraldo’
Line 152 – ‘from’
Line 158 – ‘In addition to intact genes, the T. cruzi genome…’
Line 165 – ‘may’
Line 168 – ‘meiotic’
Line 192 – I’ve not encountered ‘T. crazy’ before; something to look out for I guess…!
Line 230 – ‘difficult’
There are some typos in the references, e.g. ref 69
Reviewer 2 Report
There are some changes to the English and text to be made, as described here:
Line 1. Title. Replace 'Cruzi' with 'cruzi'.
Line 1. Title. Add a colon for punctuation following the word 'later'.
Line 5. The name of the journal Science should be in italic font.
Line 6. 'trypanosomatid' (without final letter s).
Lines 6, 24, 41. Trypanosoma not Trypanosome.
Line 6-7. Replace 'This publication is a milestone..' with 'That publication was a milestone...'.
Line 10. Replace '..before this publication..' with '..before that publication..'.
Line 12-13. Replace 'In the case, Trypanosoma..' with 'In the case of Trypanosoma..'.
Line 14-15. replace '..implementation of more advance Next- Generation, and Next-Next-Generation sequencing strategies, to this date no final..' with '..implementation of more advanced sequencing strategies, to date no final..'.
Line 21-23. The last sentence of the Abstract can be deleted.
Line 29. Triatominae should not be in italic font.
Line 34. Amend to: 'Non-vectorially transmitted infections..'.
Line 45. Chagas not Chagas's.
Line 53. Does the author mean 'Human Genome Project' to correspond with the abbreviation 'HGP'? See also the use of 'HGP' in Line 75.
Lines 67, 70. Trypanosomatidae should not be in italic font.
Line 74. Replace 'England' with 'UK'.
Line 86. Replace 'pb' with 'bp'.
Line 103. Amend to: Mucin Associated Surface Protein.
Line 104. Replace 'comprehended' with 'comprised'.
Line 106. Replace 'Emeraldo' with 'Esmeraldo'.
Line 109. Replace 'assemble' with 'assembly'.
Line 112. Amend to: '..made up of simple..'.
Line 117, 247. Replace 'Tryps' with 'trypanosomatids'
Line 146. Amend to: 'Aneuploidy and variable copy numbers of genes..'.
Line 174, Replace 'suffer ' with 'display'.
Line 192. Replace 'T. crazy' with 'T. cruzi'.
Line 194. Replace 'firsts' with 'former'.
Line 227-228. Amend to: 'its capacity'.
Line 230. Amend to: '..very difficult'.
